# Updated Understanding of Cancer as a Metabolic and Telomere-Driven Disease, and Proposal for Complex Personalized Treatment, a Hypothesis

**DOI:** 10.3390/ijms21186521

**Published:** 2020-09-07

**Authors:** Cristian Muresanu, Siva G. Somasundaram, Sergey V. Vissarionov, Luis Fernando Torres Solis, Arturo Solís Herrera, Cecil E. Kirkland, Gjumrakch Aliev

**Affiliations:** 1Research Center for Applied Biotechnology in Diagnosis and Molecular Therapies, Str. Trifoiului nr. 12 G, 400478 Cluj-Napoca, Romania; cristim23@gmail.com; 2Department of Biological Sciences, Salem University, Salem, WV 26426, USA; siva.somasundaram@salemu.edu (S.G.S.); EKirkland@salemu.edu (C.E.K.); 3The Department of Spinal Pathology and Neurosurgery, Turner Scientific and Research Institute for Children’s Orthopedics, Street Parkovskya 64-68, Pushkin, 196603 Saint-Petersburg, Russia; vissarionovs@gmail.com; 4The School of Medicine, Universidad Autónoma de Aguascalientes, 20130 Aguascalientes, Mexico; lfts99@yahoo.com.mx; 5Human Photosynthesis© Research Centre, 20000 Aguascalientes, Mexico; comagua2000@gmail.com; 6Sechenov First Moscow State Medical University (Sechenov University), St. Trubetskaya, 8, bld. 2, 119991 Moscow, Russia; 7Research Institute of Human Morphology, Russian Academy of Medical Science, Street Tsyurupa 3, 117418 Moscow, Russia; 8Institute of Physiologically Active Compounds, Russian Academy of Sciences, Chernogolovka, 142432 Moscow, Russia; 9GALLY International Research Institute, 7733 Louis Pasteur Drive, #330, San Antonio, TX 78229, USA

**Keywords:** DNA damage, cancer, DNA mutations, telomere, genetic, immune checkpoint therapy, fasting, mitochondrial metabolic reprogramming

## Abstract

In this review, we propose a holistic approach to understanding cancer as a metabolic disease. Our search for relevant studies in medical databases concludes that cancer cells do not evolve directly from normal healthy cells. We hypothesize that aberrant DNA damage accumulates over time—avoiding the natural DNA controls that otherwise repair or replace the rapidly replicating cells. DNA damage starts to accumulate in non-replicating cells, leading to senescence and aging. DNA damage is linked with genetic and epigenetic factors, but the development of cancer is favored by telomerase activity. Evidence indicates that telomere length is affected by chronic inflammations, alterations of mitochondrial DNA, and various environmental factors. Emotional stress also influences telomere length. Chronic inflammation can cause oxidative DNA damage. Oxidative stress, in turn, can trigger mitochondrial changes, which ultimately alter nuclear gene expression. This vicious cycle has led several scientists to view cancer as a metabolic disease. We have proposed complex personalized treatments that seek to correct multiple changes simultaneously using a psychological approach to reduce chronic stress, immune checkpoint therapy with reduced doses of chemo and radiotherapy, minimal surgical intervention, if any, and mitochondrial metabolic reprogramming protocols supplemented by intermittent fasting and personalized dietary plans without interfering with the other therapies.

## 1. Introduction

Cancer is a tumorigenesis process that forms a mass of cells that we call a tumor. During tumorigenesis, the cells that compose the tumor can be benign or malignant. When the cells in the tumor are normal but old, the tumor is termed benign. When the cells in the tumor are abnormal and can grow uncontrollably, the tumor is malignant. Sometimes a benign tumor can transform into a malign one if the normal old cells begin to develop abnormalities, such as DNA mutations, and grow rapidly [1,2].

Benign tumors neither metastasize nor invade local tissues. Over time, they grow and displace or compress tissues, which may cause nerve damage, reduce blood flow, or stimulate necrosis and organ damage. Or the tumor cells can transform into cancer cells and continue growing, invading tissues, and metastasizing throughout the body.

Multistage carcinogenesis is a process in which cellular alterations lead to the formation of cancer in three stages: (a) Initiation (the first cell genetic mutation); (b) promotion (a repeated division of the mutated cell); (c) progression (multiple genetic mutations and cell divisions) [3,4,5,6]. Although cancer is considered a slow-progression disease, there are exceptions, such as childhood acute lymphoblastic leukemia, which develop in a short time; however, 98% of children go into remission within weeks after starting treatment, and about 90% of these children can be cured [7].

The American Cancer Society has estimated that 1,762,450 new cancer cases were diagnosed, and 606,880 cancer deaths occurred in the United States in 2019 [8]. In 2020, 1,806,590 new cancer cases were diagnosed, and 606,520 cancer deaths are projected to occur in the United States [9].

## 2. Cancer Mechanism

Over 100 types of cancer are identified in the literature. These cancers have diverse characteristics; so, it is not reasonable to assume a single treatment plan would be effective against all of the types. Following the logic of tumorigenesis leading to carcinogenesis, we make the simplifying assumption that some form of internal cell damage must affect the benign stage, because of its natural cycle of normal cell aging and turnover, adversely. Of the possibilities for cell damage, two are linked significantly with cancer development: (a) DNA mutations; and (b) mitochondria metabolic dysfunction. We analyze these slow progression types of cancers, excluding the exceptions [7].

### 2.1. DNA Cell Mutations and Telomere Length

First, a normal cell must undergo one or more events that damage the DNA over some period of time. DNA damage and DNA mutation are distinctly different processes. DNA damage naturally occurs within the body all of the time. Still, these changes are not threatening as long as our bodies have functional DNA damage responses and recognize the DNA damage to initiate the proper cellular response [10,11].

DNA damage and DNA mutation also have different biological consequences. DNA damage is an alteration in the chemical structure of DNA, while DNA mutation is an alteration of the nucleotide sequence. Mutations result from errors during DNA replication, mitosis, and meiosis, or other types of DNA damage; therefore, in order for a normal cell to evolve into cancer, it must undergo one or more types of DNA damage [12,13,14,15].

As a consequence, we must identify causes of DNA damage: (a) genetic; (b) epigenetic. The most likely cell types that manifest DNA damage are rapidly replicating cells, because most types of DNA damage occur during the replication cycle. Unrepaired DNA damage can alter gene function or regulation of gene expression, which eventually may lead to cancer. However, unrepaired DNA damage also may accumulate in non-replicating cells, leading to cell senescence and aging. For cancer to grow, these alterations are passed on to subsequent cell generations [16,17,18,19].

Epigenetic factors that lead to DNA damage include environmental agents, such as UV light, ionizing radiation, and genotoxic chemicals that are known to be carcinogenic. However, the majority of known cancer types develop because cell replication is affected directly by the telomere length, which is maintained by the telomerase enzyme abnormal activity. Telomere length and telomerase activities are essential factors for tumor survival and carcinogenesis [20,21,22]. There is strong scientific evidence for the link between emotional stress, telomere length, chronic inflammations, alterations of mitochondrial DNA, and environmental factors [23,24,25,26,27,28,29,30,31,32,33,34,35,36]. Chronic inflammation can cause oxidative DNA damage and is one of the leading causes of obesity. Obesity can lead to cancer [37,38,39,40,41,42,43,44,45,46,47,48].

With the passage of time, the protein-based DNA repair mechanism becomes insufficient. Fewer non-replicating cells (such as the brain, muscle, and liver cells) are repaired, thus becoming senescent. Concomitant with fewer cells being repaired, more of them undergo apoptosis, especially in the presence of excess unrepaired DNA damage. However, even if the ability of DNA repair enzymes to trigger apoptosis was known to prevent cancer, other cells may become apoptosis-resistant and survive. The reduction of the DNA repair mechanism is known as DNA repair-deficiency disorder—a medical condition that can lead to accelerated age related diseases, or an increased risk of cancer. An inherited impairment in the DNA repair mechanism may also increase the risk of cancer [49,50,51,52,53].

### 2.2. Cancer Complex Mechanism Schematics

With few exceptions, cancer is a gradually emerging disease. It depends on how the environmental conditions affect the cells, internally, and the body as a whole. If the conditions are favorable, then healthy cells are adapting to the ongoing changes, but if they are not, then a group of generating factors start to affect the body. The generating factors are mainly genetic and epigenetic. Therefore, genetic factors could potentially affect DNA-strand break repair at old age. Epigenetic factors are various, and they include UV light (affecting the skin), ionizing radiation, and genotoxic chemicals (such as food mutagens, industrial chemicals, cigarette smoke, and others).

All these factors are determining and/or correlated with the emergence of cell damage. The available medical databases conclude that cancer cells do not evolve from normal healthy cells. There are two kinds of cell damages that are mostly specific to future cancer development: (a) DNA damages and (b) mitochondrial dysfunction, and they both interfere with each other (represented with a double sense arrow). DNA damages occur in both rapid replicating cells and in non/slow replicating cells. Mitochondrial dysfunction started to manifest without mitochondrial DNA mutations, leading to adenosine triphosphate (ATP) depletion and decreased ATP synthesis, which affect the extracellular matrix and promote inflammation.

Unfortunately, due to the fact that cancer cells need a lot of ATP (for maintaining a high rate of cancer metabolism), the mitochondrial reactive oxygen species (ROS) are known to trigger hypoxia-induced transcription factors which promote extracellular ATP production, followed up shortly by ATP internalization inside the cancer cells.

Oxidative stress (OS) is an internal generating factor which occurs within any tissues and cells, and it reflects the amount of an imbalance between the systemic manifestation of ROS and a biological system’s ability to readily detoxify the reactive intermediates or to repair the resulting damage. There is a multiple connection between OS, inflammation, and DNA damage. It is important to note that during DNA damage, the telomeres shorten, but during DNA mutations, after stage 1 cancer initiation, the telomeres lengthen, to ensure the survival of the cancer cell. OS also leads to obesity, which is known to promote tumorigenesis. But OS, together with the entire chain of mitochondrial dysfunction, including chronic inflammation, also favor the development of degenerative disc disease (DDD).

DDD is influenced by chronic emotional stress and generates intense chronic pain, which leads to depression. Aberrant DNA damage accumulates over time, while avoiding the natural DNA controls that otherwise repair and/or replace the rapidly replicating cells; therefore, it led to genomic instability, DNA mutations, and malignant cell formation. DNA damage also starts to accumulate in non-replicating cells, leading to senescence, (cell aging with functional deficiencies) and benign cell formation.

OS promotes telomere shortening, which in turn enhances the DNA damage, and it can also trigger mitochondrial changes, which ultimately alter nuclear gene expression. Strong evidence indicates that telomere length is affected by chronic inflammations, alterations of mitochondrial DNA, and various environmental factors. Emotional stress also influences telomere length. Cell apoptosis is known to stop cancer spreading, but many cancer cells can become apoptosis-resistant.

This stage is known as stage 1 cancer initiation (when the first cell genetic mutation occur), and/or transformation of stage 1 benign cell transformation into a cancer cell. Sometimes a malignant tumor can develop inside a larger benign tumor, therefore, it could remain undetected for a significant period of time. Both types of tumors are using telomerase activation to increase proliferation by adding new telomeres at the ends of DNA strains. However, the immune system is still able to act, and if conditions are favorable, cancer cells are marked and destroyed by white blood cells, curing the cancer.

But if it fails, then cancer continue to evolve into stage 2 cancer promotion (characterized by repeated division of the mutated cell and telomere length increasing). Usually, this is the moment when a cancer patient is seeking therapy. This vicious cycle has led several scientists to view cancer as a metabolic disease. Proposed treatments that seek to correct multiple changes simultaneously using a psychological approach to reduce chronic stress, immune checkpoint therapy with reduced doses of chemo and radiotherapy, minimal surgical intervention, if any, and mitochondrial metabolic reprogramming protocols supplemented by intermittent fasting and personalized dietary plans.

Blue arrows indicate the places these therapies may have a potentially beneficial influence in reducing a certain pathway, whenever possible. A combined version of these therapies may not cure the cancer, but they could stop cancer progression, giving time for the patient’s own biological repair mechanisms to work properly. Treating cancer is also a challenge against the passage of time. In this regard, all four proposed novel therapies, in conjunction with a less-harmful application of the old traditional classical approaches (chemotherapy, radiotherapy, and surgery), must be addressed to the cancer patient. More details have been illustrated in Figure 1.

### 2.3. Cancer and the Mitochondria Metabolic Dysfunction

As noted previously, there is a connection between obesity, chronic inflammation, and cancer. On the other hand, following the logic path, chronic inflammation may lead to oxidative stress and, therefore, DNA damage. Oxidative stress, in turn, also can trigger mitochondrial changes, resulting in mitochondrial dysfunction. Obese patients have a higher risk of cancer, due to an increase in nutrient supplies, overwhelming the Krebs cycle, and the mitochondrial respiratory chain [54,55].

Studies have shown that mitochondrial dysfunction is implicated directly in cancer development and progression [56,57]. Scientists have found that during oncogenesis, mitochondrial dysfunction occurs. This, in turn, stimulates cytosolic signaling pathways and gene expression. This process is known as retrograde signaling [58,59,60].

Another line of research has demonstrated that normal cells do not become cancerous when their nuclei are exchanged with those from cancerous cells and vice versa. Cancer cells do not become normal when their nuclei are exchanged with those from normal cells. That is, normal cells remain normal, even with a cancerous nucleus; cancer cells remain cancerous, even with a normal nucleus. These findings challenge the conjecture that cancer is due to somatic mutation. However, these findings are consistent with the hypothesis that cancer is due to mitochondrial metabolic dysfunction [61,62,63,64].

The impact of these findings is that investigations into cancer origin reasonably shift from focusing on cell nuclei to focusing on mitochondrial metabolism. Research in vitro has shown that replacing the cytoplasm of a cancer cell with the cytoplasm from a normal cell extinguishes the oncogenic phenotype. This is achieved naturally by fusing cytoplasts from the normal cell with karyoplasts present in the malignant cells [65,66,67].

A prominent advocate of the theory that cancer is a metabolic disease is Dr. Thomas Seyfried from Boston College. In his many published reports, he has argued that metabolic fermentation waste products can destabilize the tumor microenvironment, resulting in inflammation, angiogenesis, and progression [68,69]. Other researchers have confirmed that oncogenes and tumor-suppressors are associated with cancer-related changes in metabolism [70].

Other authors reports recommended that Professor Seyfried’s research findings warrant replication studies, as well as new research to test the hypothesis that cancer is a metabolic disorder [71].

In support of this hypothesis, other scientists concluded that: “[Oncogenic kinase Akt1 is activated in the nucleus under mitochondrial stress” [72,73], and “mitochondria-to-nucleus retrograde signaling plays a key role in oncogenic transformation of breast epithelial cells” [74,75].

Many researchers agree that mitochondrial dysfunction contributes to oncogene-induced senescence [76,77] with dysfunctional mitochondria accumulating around the cell nucleus. This is accompanied by oxidative DNA damage and suggests that mitochondrial dysfunction may represent a means for oncogene-induced senescence [76,78,79]. Mitochondrial dysfunction plausibly underlies aging-associated brain degeneration, and the mitochondrial damage correlates with increased intracellular production of oxidants and pro-oxidants [80].

In order to demonstrate that cancer onset and progression is based on mitochondria metabolism instead of the DNA of the cell nucleus, research is needed that reprograms mitochondrial metabolism instead of killing the cancer cells. When our bodies are healthy, cancerous cells naturally undergo apoptosis. If that does not happen, then the cancer cells usually are marked by the immune system and destroyed by white blood cells. Consequently, cancer only develops when the body becomes unable to perform one or both of these natural physiological processes.

The presumptive benefits from chemotherapy and radiotherapy are that, after the treatment, the patient’s immune system is able to identify and destroy the remaining cancer cells. The inherent problem with these treatments is that the patient’s immune system may be so severely diminished that even if the patient does not die of cancer, other diseases have the opportunity to flourish relatively unchecked.

## 3. Cancer Cell Behavior

Logically, we should focus our attention on generating factors that affect telomere length in carcinogenesis and mitochondria metabolic dysfunction. There are many debates concerning novel telomere therapies. It is well known that longer DNA telomere is associated with longer life span, greater genomic stability, and better DNA repair efficiency, and cell replacement.

However, when DNA cell damage becomes advanced, one or more of three cellular maintenance systems are failing: (a) DNA repair, (b) The cell-cycle braking system, and/or (c) Telomere loss. Usually, they fail simultaneously, albeit gradually. There is a bidirectional connection between telomere shortening associated with aging, early senescence, and premature cell death, and mitochondrial dysfunction that produces reactive oxygen species (ROS), leading to oxidative damage of cellular constituents. Viewing the process as a vicious feedback loop may offer a broader view for understanding the pathophysiology of these diseases [81,82,83,84,85].

It has been proposed that patients without cancer would benefit from therapies resulting in telomere extension. While no one has suggested telomere length alone is responsible for preventing cancer, telomere length is associated with the body’s ability to stop tumorigenesis before it gains a foothold and progresses to carcinogenesis [85,86]. However, any intervention involving telomerase enzyme activity in vivo in humans may pose unforeseen, possibly adverse side effects. In addition, such therapy is not encouraged for patients who already have cancer. Therefore, the current stage of research has not yet passed animal model studies.

Telomere loss is associated with the earliest stages of cell damage and oxidative stress. In cancer cells, telomere growth is associated with fermentation metabolism. Cancer cells appear to reprogram their metabolism, based on the increased glucose uptake and fermentation of glucose to lactate, even in the presence of competently functioning mitochondria. This is known as the Warburg Effect. However, the functional basis of this effect remains unclear [87,88]. Some cancer cells cannot survive without glutamine, and they change glutamine to lactate [89,90,91,92]. Actually, we have two kinds of telomere problems: (a) telomere shortens, in aged cells, and (b) telomere lenghten in cancer cells. These findings make any cancer treatment more complicated to explain and apply.

Nonetheless, Warburg and his colleagues proposed that dysfunctional mitochondria trigger an aerobic glycolysis mechanism [93,94]. The fermentation mechanism can occur even in the presence of the oxygen, and thus, Warburg named it aerobic glycolysis [95,96,97]. However, the Warburg Effect remains poorly understood, although several hypotheses have been proposed to explain it. Cancer cells exhibit ongoing metabolic oxidative stress compared with normal cells [98,99,100]. They induce many types of DNA damage, including gene overexpression, DNA single- or double-strand breaks, erroneous arrangement of the DNA sequence, oncogene expression, and DNA lesions [101].

These discoveries highlight the complexity of cancer cell behavior, vitiating many proposed therapies, and contributing to adverse side effects. Many in vitro findings have failed to be replicated in vivo. Therefore, we propose a new type of medical intervention to address cancer using the body’s own defenses.

## 4. Autophagy as a Potential Treatment

Autophagy (or autophagocytosis) is a potential treatment based on naturally occurring autorecovery mechanisms in the human body. The body can correct both mitochondrial dysfunction and adverse telomerase enzyme activity naturally. This proposition is consistent with trends in cancer immune checkpoint therapy that seek to enhance the role of cytotoxic T-lymphocyte-associated protein (CTLA-4), programmed cell death protein 1 (PD-1), and programmed cell death protein ligand 1 (PD-L1) [102].

Research has shown that starvation induces non-selective autophagy in yeast. Autophagy [103] can be induced in many types of cells in response to nutritional deficiencies. The autophagic bodies accumulate in the phagolysosomal stage of vacuoles, enabling the disposal of degraded proteins [104,105,106,107]. Autophagy is a naturally regulated mechanism that removes unnecessary or dysfunctional components from cells [108]. This is part of a precise, controlled recycling or regeneration process throughout the body [109].

However, autophagy as a tumor cell suppressor remains controversial. Some cancer studies have shown that “autophagy is a survival pathway activated during nutrient deprivation and other stresses”, therefore, “it is reasonable to think that autophagy can function as a tumor cell survival mechanism activated after cancer treatment” [110]. However, when cancer cells become apoptosis-resistant and if there is an autophagic deficiency, then “the combined impairment of apoptosis and autophagy promotes necrotic cell death in vitro and in vivo”, therefore, to avoid this situation, “autophagy may function in tumor-suppression by mitigating metabolic stress and, in concert with apoptosis, by preventing death by necrosis” [111].

### 4.1. Main Autophagic Mechanisms

Autophagy is an advanced recycling mechanism that is used by the body for several operations, including (but not limited to): (a) Eliminating damaged or dysfunctional mitochondria, known as mitophagy [112]; (b) eliminating or degrading the lipids or lipid structures called lipid droplets, known as lipophagy [113]; (c) engulfing cytoplasmic material into lysosomes, known as microautophagy [114]; (d) eliminating or recycling the damaged cell organelles or unused proteins, known as macroautophagy [115]; and (e) chaperoning autophagy for the selection of soluble cytosolic proteins, targeted to lysosomes and translocated across the lysosome membrane for degradation [116].

Related genes and specific proteins regulate these autophagy mechanisms. The main functions of autophagy mechanisms are: (a) Nutrient starvation (for degrading unnecessary proteins down to basic amino acids and then using these amino acids for the synthesis of proteins) [117,118]; (b) Repair mechanism (for degrading or recycling damaged organelles, cell membranes, and other proteins, preventing premature aging) [119]; (c) Apoptosis or programmed cell death (for eliminating or degrading the damaged cells which are unable to fulfill their functions; and autophagic activity in dying cells) [120,121]; (d) Other mechanisms, (such as xenophagy, for degradation and recycling infectious particles and pathogens) [122].

### 4.2. Autophagy and Cancer Treatment

Although there are many interacting factors associated with cancer and cancer therapies, the autophagic mechanisms are naturally occurring in the human body. If researchers are successful in applying them, the results would be very encouraging. However, depending upon environmental conditions, autophagy may induce tumor-suppression when the Beclin1 gene is overexpressed [123], or it can promote tumor cell survival after the cancer cells have been hit with radiation [124], thus making radiotherapy ineffective in several types of cancer, as well as making cancer cells resistant to radiation.

With conservative treatment, autophagy is more likely to inhibit tumor growth than promote it [125], but other studies have revealed that increased autophagy promotes tumor cell survival and growth [126,127]. Other scientists also confirmed that in pancreatic cancer, the genetic inhibition of autophagy inhibited tumor growth [128]. The only logical follow-up would be the practical application of the autophagic mechanism for the whole body, which can be achieved by fasting.

Researchers have proposed fasting as an adjuvant for various cancer therapies [129,130]. In healthy human bodies, autophagy helps prevent cancer development by maintaining normal cell homeostasis, and by removing oncogenic protein substrates, toxic un-degraded proteins, and damaged organelles. There has been considerable research on fasting and dietary restrictions for enhancing cancer therapy, although many experiments have been conducted only on mice [131]. Moreover, over 100 published studies have tested the influence of diet on anti-cancer immune responsiveness [132], the link between nutrition, inflammation, and cancer [133], calorie restriction and fasting in delaying tumorigenesis [134] and even preventing cancer [135], or even the ketogenic diet, which was associated with improvements in cancer therapy [136], as well as many others.

However, applying fasting in a clinical trial poses many challenges because when mitochondrial dysfunction is detected, prolonged fasting is not recommended: Prolonged fasting can lead to anemia—it may weaken the immune system, exacerbate some liver and kidney problems, or lead to an irregular heartbeat. To avoid such problems, researchers have proposed intermittent fasting, also known as time-controlled fasting or short-term fasting. These therapeutic strategies have been effective in limiting mitochondrial damage and metabolic disturbances induced by modern Western diets or aging [137]. Short-term fasting protects the individual from toxicity and reinforces the stress resistance of healthy cells, while tumor cells become even more sensitive to toxins and chemotherapeutic agents [138].

Controlled fasting means that the quantity of food is reduced without resulting in malnutrition. In order to avoid psychological stress during fasting, scientists and dieticians have proposed several means to increase food restriction, such as intermittent calorie restriction, calorie restriction, intermittent fasting, water fasting, and dry fasting.

Fasting is one of the oldest known therapies, and is an integral component of various spiritual traditions. However, fasting is not without its detractors. Some scientists argue there is no clear indication or strategy for recommending fasting before or after chemotherapy [139]. There is limited research on the short-term effects of fasting with respect to reducing toxicity [140]; and, among other things, the media attention given to fasting overwhelms the scientific bases in oncology [141]. Admittedly, these are valid points that highlight the need for advanced controlled research studies to determine the optimal use of fasting as an adjuvant to cancer therapy [129,130], as noted above, not as cancer therapy per se.

Nevertheless, there are too many positive indications of the value of fasting with respect to autophagy in cell maintenance and body health. Of course, there are many potential confounding factors in applying fasting because it may be associated with radical changes in lifestyle, exercise, and diet. Reasonably, we expected that the benefits would accrue unevenly according to perhaps the normal curve, with some individuals having weak, temporary, or inconsistent results.

In addition, high levels of stress are associated with telomere shortening, which would be an adverse side effect. Accordingly, personal-psychological response to stress is important. High levels of stress have been reported to contribute to premature aging and earlier onset of diseases associated with aging [142]. That suggests that fasting may not be helpful if the patient feels fasting is an additional stress factor in a life already stressed by cancer and its many sequelae. Furthermore, no diet has been shown to cure cancer, therefore, it is not expected that fasting alone can cure cancer.

It is also important to add that proteasome inhibitors (like Bortezomid or the histone deacetylase (HDAC) inhibitor—such as Panobinostat) may induce autophagy in the plasma cell for treating multiple myeloma [143,144].

## 5. Novel Cancer Immune Checkpoint Therapy

With their Nobel Prize winning research, Allison and Honjo argued for a new approach to cancer therapy with very promising outcomes using the inhibition of negative immune regulation by inhibiting the CTLA4 and PD1 immune checkpoints that allow T cells to eradicate cancer cells [145]. This discovery offers the potential to enhance rates of cure for many types of cancer, including forms of cancer that are resistant to chemotherapy or radiotherapy. The effect is achieved by boosting the anti-tumor immune responses that spontaneously arises in many patients [146]. This research focuses on understanding the mechanisms of action and improving cancer therapies, while simultaneously reducing side effects. Therefore, so-called checkpoint therapy has improved cancer treatment and has fundamentally changed the way we view how cancer can be managed [147].

This therapeutic approach does not exclude the use of other therapies. The combinational therapy between immune checkpoint therapy and other cancer therapies could lead to a higher rate of success in curing many forms of cancer [148]. For example, using chemotherapy as a pre-treatment in combination with immune checkpoint therapy might enhance the antigenicity and immunogenicity of tumors by promoting adaptive immune responses [149].

The immunogenic cell death and systemic tumor response have been observed during localized radiotherapy [150], which can be combined with immune checkpoint therapy. Reasonably, further research studies are needed to establish and ensure no harm to patients, especially because adverse autoimmune effects have been observed in some cases during checkpoint blockade therapies [151].

With regard to the role of growth factors role in cancer, significant results were achieved when PD-1/PD-L1 blockade was combined with inhibition of transforming growth factor beta (TGF-β) [152]. The monomorphic major histocompatibility complex class I-related gene protein (MR1) can be used in targeting and destroying cancer cells, while being nonharmful for noncancerous cells [153]. Lab tests have shown that T-cells enhanced with a special T-cell receptor (TCR) can destroy a wide range of cancer cells because the newly discovered TCR can recognize many types of cancer with the help of MR1 (which does not vary in human populations) [153,154].

Admittedly, theoretic benefits and in vivo findings with lab animals, do not guarantee any benefit with humans. Aside from the multiple orders of magnitude in the complexity of humans versus animals, the human mind has unlimited ability to influence the way the body acts and reacts. In addition, cancer cells are known to adapt to their environment and become perhaps 100 times more likely to genetically mutate than regular cells [155].

Therefore, many promising therapies are susceptible to becoming less effective over time. Given the potential that cancer cells become “mutator” cells, chemo therapies and radiotherapies may never achieve a complete cure [155]. While it may become possible one day to reprogram cancer cells to become “normal” cells again, there is a long-term research and development process ahead to achieve such a goal [156].

## 6. Mitochondrial Metabolic Reprogramming in Cancer Therapies

The premise that the cancer cells’ mitochondria are damaged or otherwise defective has been called into question by the behavior of cancer cells themselves. A hallmark of cancer is rapid cell growth and proliferation. Clearly, the energy requirements to sustain growth must be met, or else carcinogenesis would falter. Logically, if the cancer cells’ mitochondria are damaged or defective, one must raise the question of how they can meet the needs for rapid growth that clinical evidence has revealed is commonplace in cancer.

Current research has improved our knowledge of the means by which aerobic glycolysis and other metabolic alterations in cancer cells can sustain the anabolic requirements for cell differentiation and proliferation. A recent reinterpretation of Warburg [93] concludes that the cancer cells’ mitochondria cannot be defective per se because of their ability to maintain the complex oxidative phosphorylation process to synthesize adenosine triphosphate (ATP) [157,158].

Thus, instead of being defective, the mitochondria appear to have been reprogrammed. In proliferating cells, the production of ATP by oxidative phosphorylation appears secondary to using mitochondrial enzymes and the synthesis of anabolic precursors. The transformation results from mutations in proto-oncogenes and tumor-suppressors. The metabolites themselves alter cell signaling and responses. Reprogramming of mitochondrial citrate metabolism, which is found only in mitochondria, is a central factor in oncogenic activity [157]. This finding, in turn, suggests a new approach for modeling new protocols for regenerative therapies [159,160]. It appears that genetic and epigenetic factors are involved in reprogramming, which is communicated between mitochondrial and nuclear genomes [161].

Hopefully, new insights will encourage the development of new therapies that capitalize on regenerative capabilities, which would be especially important for effective, timely treatment of highly aggressive or treatment-resistant tumors and cancers. Mitochondrial reactive oxygen species are known to trigger hypoxia-induced transcription [162]. Hypoxia causes the activation of hypoxia-inducible factor 1 (HIF-1) and accumulation of extracellular adenosine. Both factors are important in supporting tumor growth as HIF-1 controls angiogenesis, and adenosine exerts a profound immunosuppressive activity, thus protecting the tumor from inflammatory cells [163]. Recent data show that solid tumors have a gradient of adenosine concentration from the center to the periphery, higher than the surrounding healthy tissue [164].

Other research in mitochondrial dysfunction has revealed that mitochondria are vitiated significantly in Alzheimer’s disease [165]. Evidence supports the theory that oxidative stress, mostly due to reactive oxygen species, induces mitochondrial damage [166], and initiates mitochondrial failure that is an initiating factor of Alzheimer’s disease [167].

## 7. Cancer and Growth Factors Therapies

Growth factors stimulate cellular growth, proliferation, healing, and cellular differentiation [161]. In adverse cases, these same growth factors stimulate tumor growth and cancer cell proliferation. However, while growth factors per se are not a cause of cancer, some types of cancer cells manage to synthesize growth factors to which they are responsive [168]. Eventually, these growth factors promote membrane disruption and tumor progression, as well as the growth of blood vessels to supply the tumor. This enables cancers to acquire vascular endothelial growth factors to promote blood vessel growth [169].

Unfortunately, various growth factors therapies have failed to show positive results. These problems were reviewed previously [170]. The inefficiency of growth factor therapies, explained for degenerative disc disease, is applicable to growth factors therapies for cancer and other incurable diseases, simply because for curing these diseases is needed a continuous and uninterrupted supply of the optimal growth factors mixture and is not otherwise possible.

While growth factors are important for carcinogenesis, at the same time, the mind can control bodily mechanisms to produce growth factors from seminal secretions that help repair injured or aged cells and tissues. Importantly, growth factors do not appear to benefit cancer cells. That means that it is more likely that autologous growth factors may induce cytotoxic T cells or natural killer cells to phagocytose the cancer cells, rather than normal cells [171].

## 8. Nutrition as an Adjuvant in Cancer Therapy

Although there is limited information on the direct impact of nutrition on cancer, there is evidence that nutrition generally affects basic cellular and molecular processes that are fundamental to oncogenesis, carcinogenesis, and progression. For example, exposure to alcohol and processed meats are likely causal factors [172]. Review papers report that there exists a potential role of diet in certain cancers. Sadly, nothing has been conclusive. Also, many findings are confounded by lifestyle, psychological health, stress responses, and many other uncontrolled intervening factors [173].

However, some scientists are convinced that carcinogenic and mutagenic factors are prevalent in the Western diet today [174]. Therefore, nutrition is increasingly important in preventing or promoting carcinogenesis. In this regard, the diet may be used to enhance cancer therapies, either as an important adjuvant or in combination with other therapies, but without compromising the ongoing treatments aiming at inducing ROS production in cancer cells. In this regard, researchers proposed a new form of autophagy, known as autophagy modulation, as a novel potential therapy for treating some of the malignancies [129,175,176,177]. We presented that caution should be exercised when prescribing dietary antioxidant supplements for certain cancer chemotherapies [178].

It has been recommended that cancer patients with low immunoreactivity and liver problems are helped by diets, including raw fruits and vegetables, as well as raw liver. These foods provide oxidizing enzymes that help the liver recover [174]. Other studies report that cancer is a nutritionally responsive disease that is amenable to nutrient-based and immunotherapeutic treatments [179]. Some scientists have suggested that nutrition can act help prevent cancer, such as the various studies that have shown the cancer-preventing value of dietary fiber, as well as the benefits of diets rich in antioxidants [180].

There are indications that functional foods enhance health and reduce the risk of many diseases. There are reports of an association between functional foods and suppressing cancer [181]. Another diet therapy approach is the mixed fasting and ketogenic diets that restrict macronutrient intake [182]. Cancer cells can grow in the absence of growth factors, but they do not respond to nutrient deprivation the same way as healthy cells: They continue to proliferate even when nutrients are scarce. Dietary restrictions are valuable in reducing toxicities that accompany some cancer therapies [182].

## 9. The Role of Chemotherapy and Radiotherapy in Cancer Treatment and Their Outcome

It is undeniable that the classical known chemotherapies and radiotherapies have played an important role in treating cancer, in the past 60 years for chemotherapy, and even more for radio therapy. Chemotherapy has been proposed to be used in order to interfere with cancer cell division, but cancer cells are behaving very differently when cytotoxic agents are introduced into the bloodstream. 

Therefore, chemotherapy has limitations, and obviously, a combinational therapy, increases the efficacy of treatment, such as the one we have proposed in this paper. Recent studies revealed that microRNAs (miRNAs) are known to suppress malignant development in a normal cell [183].

Therefore, miRNAs may show a potential application in treating cancer, by restoring tumor-suppressive miRNAs while targeting the oncogenic miRNAs as well [184,185]. In this regard, it is plausible to highlight the role of chemotherapeutic agents in conjunction with miRNAs for cancer treatment, as a novel strategy [186]. Earlier studies have revealed that oxygen and nutrition deprivation in actively growing tumors lead to cancer cellular adaptation towards survival. During this adaptive process, mitochondria increase ROS - which in turn activates signaling pathways, such as hypoxia-inducible factor 1-alpha (which is responsible for cancer cell survival). Thus, the mitochondrial origin of ROS signaling may be beneficial in developing novel chemotherapy [187].

However, it is also undeniable that the side effects of chemotherapy, such as myelosuppression, immunosuppression, and others, lowers the five years survival rate on most treated cancer patients; therefore, it is not unusual and unprecedented, the use of a combination of therapies, simultaneously targeting multiple generating factors of cancer development, in order to ameliorate or even stop cancer progression.

Radiotherapy is only useful for those cancers which are localized in one single area of the body, but there are known cases of cancer patient’s survivors, which developed secondary malignancies after radiotherapy [188,189]; however, there are debates because the incidence of radiation, viewed as a major cause for secondary malignancies, is difficult to estimate.

## 10. Discussions and Conclusions for a Complex Multilevel Personalized Treatment

It is undeniable that the causes and treatments of cancer are invaluable, and research must be extended. This review summarizes an extensive survey of the PubMed database. There is no simple solution to cancer prevention or therapy because it may be the case that we need a personalized protocol for each type of cancer, according to the patient’s personality and lifestyle. Gathering the psychological, demographic, environmental, and medical profiles to combine with the cancer diagnosis and treatment is a daunting task from a holistic, clinical perspective. However, the basic features of a clinical protocol suggest the following key factors:
-Obtain an accurate history of the patient’s dramatic and stressful events during his or her lifetime and advanced psychological counseling to evaluate ways to make the patient aware of the consequences of chronic stress and means of coping with and, ideally, discontinuing current stress mechanisms.-Utilize immune checkpoint therapy with chemotherapy, radiotherapy, and surgery performed with minimal aggressiveness toward the cancer itself in order to avoid cancer resistance and to reduce the danger of tumor(s) threatening vital organs. Radiotherapies and chemotherapies should be used in such a way to minimize the impact on the immune system.-Use intermittent fasting, before and after medical interventions, but without interfering with the other proposed therapies.-Employ the mitochondrial metabolic reprogramming protocol, according to the patient’s medical conditions and identifying the oncogenic metabolites.-If the patient’s cancer have been caused or correlated with a viral/bacterial chronic infection, [190,191] then additional therapies may be needed. Also, there are other bacteria which could be used to treat cancer [192], therefore, the complex personalized cancer therapy should take into consideration all possible choices.-Provide personalized dietary restrictions to maintain the best metabolic equilibrium for avoiding cancer reoccurrence or relapse.


In addition, biological transformations controlled by the mind [193] might be recommended, but only after the complete confirmed absence of cancer cells in the body. This restriction is to avoid interactions between growth factors and any remaining cancer cells. Obviously, this is a very difficult task. We are confident that future developments will make these methods more accessible for both patients and physicians.

## Figures and Tables

**Figure 1 ijms-21-06521-f001:**
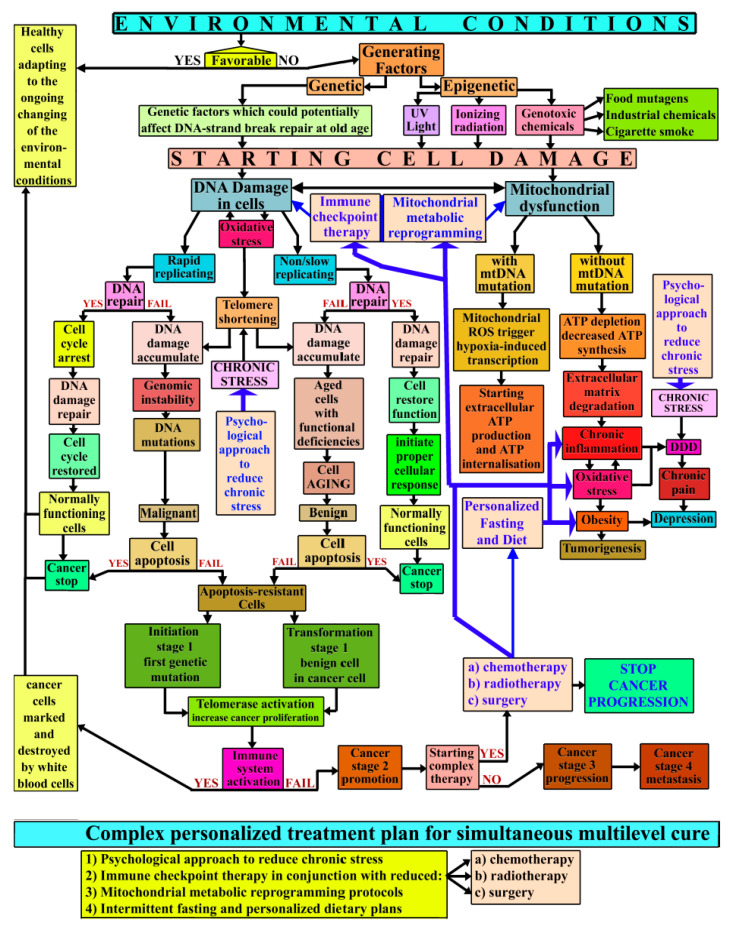
Cancer’s complex mechanism chart.

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
