# Peer review of "Updated Understanding of Cancer as a Metabolic and Telomere-Driven Disease, and Proposal for Complex Personalized Treatment, a Hypothesis"

_ijms, 2020, doi:10.3390/ijms21186521_

Round 1
Reviewer 1 Report
The Review “Updated Understanding of Cancer as a Metabolic and Telomere-Driven Disease, and Proposal for Complex Personalized Treatment, a Hypothesis” by Muresanu is an excellent written summarize about current understanding of cancer, the meaning of telomers and metabolism for this disease class. It should be published.
One comment about autophagy as cancer treatment: Proteasome inhibitors like Bortezomib or the histone deacetylase (HDAC) inhibitor Panobinostat induce autophagy in the plasma cell disease Multiple Myeloma.
typo: line 137: OS promotes
Author Response
We thank you very much for your kind appreciation and following your suggestion, we have enhanced the paper by adding on subchapter 4.2. Autophagy and Cancer Treatment the following sentence in blue color:
It is also important to add that proteasome inhibitors, like Bortezomid or the histone deacetylase (HDAC) inhibitor, such as Panobinostat, may induce autophagy in the plasma cell for treating Multiple Myeloma [143,144].
and the corresponding reference sources:
- Cea, M.; Cagnetta, A.; Gobbi, M. et al. New insights into the treatment of multiple myeloma with histone deacetylase inhibitors. Curr Pharm Des 2013; 19(4): 734-44, PMID: 23016853.
- Andreu-Vieyra, C.V.; Berenson, J.R. The potential of panobinostat as a treatment option in patients with relapsed and refractory multiple myeloma. Ther Adv Hematol 2014; 5(6): 197-210, https://doi.org/10.1177/2040620714552614.
Also, typo at line 137 have been corrected.
Reviewer 2 Report
This manuscript reviews the complexity tumorigenesis with focus on metabolic and telomerase. It provided a very thorough summary of the involvement of mitochondrial dysfunction during cancer initiation and progression, suggesting DNA damage and subsequent ROS production in mitochondria is the cause of cancer, therefore, means to decrease ROS will be essentially benefit to cancer patients. The review however omits the important facts, such as most of the breakthrough chemotherapeutics and radiotherapies that are benefit to patients right now are mediated at least partially by generating massive ROS and induce programmed cell death in cancer cells. Therefore, during this type of treatment, suggesting patients to consume antioxidative diet may compromise ongoing treatment aiming at inducing ROS production in cancer cells.
Elevated rates of reactive oxygen species (ROS) have been detected in almost all cancers, where they promote many aspects of tumor development and progression. However, reactive oxygen species (ROS) as a group of highly reactive molecules is also important signaling molecules, moderate levels of ROS are required for several cellular functions, including gene expression. Therefore, a delicate balance of intracellular ROS levels is required for normal cell function.
The article could be improved by dissecting mitochondria/ROS function during early tumorigenesis process and during cancer treatment to provide a full picture.
Author Response
We agree with your suggestions, and according to that we enhanced the paper by adding the following texts in red color:
On the abstract, lines 34 and 35 the new added text is:
dietary plans without interfering with the other therapies.
We have provided extensive explanations during the main article text.
On chapter 8. Nutrition as an Adjuvant in Cancer Therapy, lines 421, 422 and 423 we added the following text:
in combination with other therapies, but without compromising the ongoing treatments aiming at inducing ROS production in cancer cells. In this regard, researchers proposed a new form of autophagy, known as autophagy modulation as a novel potential therapy for treating some of the malignancies [175-178].
and the corresponding reference sources:
- Levy, J.M.M.; Towers, C.G.; Thorburn, A. Targeting autophagy in cancer. Nat Rev Cancer 2017; 17(9): 528-42, https://doi.org/10.1038/nrc.2017.53.
- Galluzzi, L.; Bravo-San Pedro, J.M.; Levine, B.; Green, D.R.; Kroemer, G. Pharmacological modulation of autophagy: therapeutic potential and persisting obstacles. Nat Rev Drug Discov 2017; 16(7): 487-511, https://doi.org/10.1038/nrd.2017.22.
- Fulda, S. Autophagy in Cancer Therapy. Front Oncol 2017; 7: 128, https://doi.org/10.3389/fonc.2017.00128.
- Antunes, F.; Erustes, A.G.; Costa, A.J.; et al. Autophagy and intermittent fasting: the connection for cancer therapy?. Clinics (Sao Paulo) 2018; 73(suppl 1): e814s, https://doi.org/doi:10.6061/clinics/2018/e814s.
In order to cover the other issues about the use of chemotherapy we also included a supplemental chapter 9. The Role of Chemo- and Radio- therapy in Cancer Treatment and Their Outcome in order to enhance the importance of the classical well-known chemo- and radio- therapies, in order to avoid any misleading regarding the benefits and outcomes of these therapies and we hope that this new chapter will highlight the undeniable necesity for these therapies to be used. We only enhanced the benefit of using them in conjunction with other novel therapies which are now currently under development. This association could be in the benefit of the patients, by reducing the side effects which cannot be ignored.
and the corresponding reference sources:
- Iorio, M.V.; Croce, C.M. MicroRNAs in cancer: small molecules with a huge impact. J Clin Oncol 2009; 27(34): 5848-56, https://doi.org/10.1200/JCO.2009.24.0317.
- Zhou, K.; Liu, M.; Cao, Y. New Insight into microRNA Functions in Cancer: Oncogene-microRNA-Tumor Suppressor Gene Network. Front Mol Biosci 2017; 4: 46, https://doi.org/10.3389/fmolb.2017.00046.
- Gandellini, P.; Profumo, V.; Folini, M.; Zaffaroni, N. MicroRNAs as new therapeutic targets and tools in cancer. Expert Opin Ther Targets 2011; 15(3): 265-79, https://doi.org/10.1517/14728222.2011.550878.
- Chakraborty, C.; Sharma, A.R.; Sharma, G.; Sarkar, B.K.; Lee, S.S. The novel strategies for next-generation cancer treatment: miRNA combined with chemotherapeutic agents for the treatment of cancer. Oncotarget 2018; 9(11): 10164-74, https://doi.org/10.18632/oncotarget.24309.
- Kamran, S.C.; Berrington, de Gonzalez A.; Ng, A.; Haas-Kogan, D.; Viswanathan, A.N. Therapeutic radiation and the potential risk of second malignancies. Cancer 2016; 122(12): 1809–21, https://doi.org/10.1002/cncr.29841.
- Dracham, C.B.; Shankar, A.; Madan, R. Radiation induced secondary malignancies: a review article. Radiation Oncol J 2018; 36(2): 85–94, https://doi.org/10.3857/roj.2018.00290.
On chapter 10. Discussions And Conclusions For A Complex Multilevel Personalized Treatment we added at line 473 and 473 an enhanced sentence:
Use intermittent fasting, before and after medical interventions, but without interfering with the other proposed therapies.
1. The authors agree with the reviewers point of view on ROS. One of our research published in Cancer Research in 2002 presented that certain chemotherapies increases the ROS and induces apoptosis in breast cancer in vitro and invivo. In addition, the dietary supplement curcumin inhibits the chemotherapeutic effect due to its antioxidant properties. Hence we agree with the reviewer comment "Therefore, during this type of treatment, suggesting patients to consume antioxidative diet may compromise ongoing treatment aiming at inducing ROS production in cancer cells." We presented that the caution should be exercised when prescribing dietary antioxidants supplements for certain cancer chemotherapies. We added Reference: Cancer Res. 2002 Jul 1;62(13):3868-75.
2. The authors also thank the reviewer for highlighting the role of ROS and mitochondria. It has been established that in the earlier studies that oxygen and nutrition deprivation in actively growing tumors lead to cancer cellular adaptation towards survival. During this adaptive process, mitochondria increase ROS which in turn activate signaling pathways such hypoxia inducible factor 1-alpha which is responsible for cancer cell survival. Thus mitochondrial origin of ROS signaling may be beneficial in developing novel chemotherapy. We also added reference: Clin Cancer Res. 2007; 13(3): 789-94. doi: 10.1158/1078-0432.CCR-06-2082. All new references have been inserted accordingly and the final reference list have been updated and renumbered.
We would also wish to highlight that our paper proposes a hypothesis which could stimulate discussions and debates with other scientists, without pretending that this is ultimate solution for cancer therapy. As we mentioned in chapter 10, "There is no simple solution to cancer prevention or therapy because it may be the case that we need a personalized protocol for each type of cancer, according to the patient’s personality and lifestyle."
We address many thanks to reviewer writing the review form 2 for the comments and suggestions and hopefully this new enhanced version of our paper is covering all the issues.
Round 2
Reviewer 2 Report
The authors made changes to tone down the benefits of antioxidants for cancer patients, and drive the paper more towards stimulation of scientific discussion, these changes do improve the manuscript.